# Untargeted Metabolomics Identifies Potential Hypertrophic Cardiomyopathy Biomarkers in Carriers of *MYBPC3* Founder Variants

**DOI:** 10.3390/ijms24044031

**Published:** 2023-02-17

**Authors:** Mark Jansen, Maike Schuldt, Beau O. van Driel, Amand F. Schmidt, Imke Christiaans, Saskia N. van der Crabben, Yvonne M. Hoedemaekers, Dennis Dooijes, Jan D. H. Jongbloed, Ludolf G. Boven, Ronald H. Lekanne Deprez, Arthur A. M. Wilde, Judith J. M. Jans, Jolanda van der Velden, Rudolf A. de Boer, J. Peter van Tintelen, Folkert W. Asselbergs, Annette F. Baas

**Affiliations:** 1Department of Genetics, University Medical Centre Utrecht, Utrecht University, 3584CX Utrecht, The Netherlands; 2Netherlands Heart Institute, 3511EP Utrecht, The Netherlands; 3Department of Physiology, Amsterdam UMC, Vrije Universiteit Amsterdam, Amsterdam Cardiovascular Sciences, 1081HZ Amsterdam, The Netherlands; 4Department of Cardiology, University Medical Centre Utrecht, Utrecht University, 3584CX Utrecht, The Netherlands; 5Institute of Cardiovascular Science, Faculty of Population Health Sciences, University College London, London WC1E 6DD, UK; 6Department of Genetics, University Medical Centre Groningen, University of Groningen, 9713GZ Groningen, The Netherlands; 7Department of Human Genetics, Amsterdam UMC, University of Amsterdam, 1105AZ Amsterdam, The Netherlands; 8Department of Clinical Genetics, Radboud University Medical Centre, 6525GA Nijmegen, The Netherlands; 9Heart Centre, Department of Cardiology, Amsterdam UMC, University of Amsterdam, 1081HZ Amsterdam, The Netherlands; 10Department of Cardiology, University Medical Centre Groningen, University of Groningen, 9713GZ Groningen, The Netherlands; 11Health Data Research UK and Institute of Health Informatics, University College London, London NW1 2DA, UK

**Keywords:** hypertrophic cardiomyopathy, *MYBPC3*, biomarkers, metabolomics

## Abstract

Hypertrophic cardiomyopathy (HCM) is the most prevalent monogenic heart disease, commonly caused by pathogenic *MYBPC3* variants, and a significant cause of sudden cardiac death. Severity is highly variable, with incomplete penetrance among genotype-positive family members. Previous studies demonstrated metabolic changes in HCM. We aimed to identify metabolite profiles associated with disease severity in carriers of *MYBPC3* founder variants using direct-infusion high-resolution mass spectrometry in plasma of 30 carriers with a severe phenotype (maximum wall thickness ≥20 mm, septal reduction therapy, congestive heart failure, left ventricular ejection fraction <50%, or malignant ventricular arrhythmia) and 30 age- and sex-matched carriers with no or a mild phenotype. Of the top 25 mass spectrometry peaks selected by sparse partial least squares discriminant analysis, XGBoost gradient boosted trees, and Lasso logistic regression (42 total), 36 associated with severe HCM at a *p* < 0.05, 20 at *p* < 0.01, and 3 at *p* < 0.001. These peaks could be clustered to several metabolic pathways, including acylcarnitine, histidine, lysine, purine and steroid hormone metabolism, and proteolysis. In conclusion, this exploratory case-control study identified metabolites associated with severe phenotypes in *MYBPC3* founder variant carriers. Future studies should assess whether these biomarkers contribute to HCM pathogenesis and evaluate their contribution to risk stratification.

## 1. Introduction

Hypertrophic cardiomyopathy (HCM) is characterised by left ventricular hypertrophy not explained by abnormal loading conditions and may lead to left ventricular outflow tract (LVOT) obstruction, sudden cardiac death, or heart failure [1]. Genetic variants often underlie HCM, most frequently affecting the *MYBPC3* gene [2]. This gene encodes cardiac myosin-binding protein C (cMyBP-C), an important regulator of cardiomyocyte contraction and relaxation [3].

Previous studies in the Dutch HCM population observed three pathogenic *MYBPC3* founder variants, c.2373dupG p.(Trp792*fs*), c.2827C > T p.(Arg943*), and c.2864_2865delCT p.(Pro955*fs*), that together accounted for up to 35% of HCM cases [4]. A fourth founder variant, c.3776delA p.(Gln1259*fs*), has since been identified. These variants similarly result in truncated mRNA and absent truncated cMyBP-C protein, leading to reduced total cMyBP-C (i.e., haploinsufficiency) [5] and impaired cardiomyocyte function [3]. Consequently, these variants equally impact the prognosis of carriers [6].

Clinical severity in HCM is highly variable and penetrance in pathogenic variants carriers is incomplete [6,7,8]. Prediction of HCM development and adverse cardiac events remains limited, urging the identification of novel predictors of HCM progression and its underlying mechanisms. Previous studies showed destabilisation of myosin energy-conserving states and decreases in myocardial efficiency due to HCM-causing genetic variants [3,9]. Accordingly, metabolic changes have been demonstrated in cardiac tissue of HCM patients [10,11,12]. If replicated in plasma, metabolites from affected metabolic pathways can potentially function as biomarkers for HCM development and progression. This may lead to the discovery of clinically usable prognostic biomarkers or potential treatment targets. Relevant to this purpose, the advent of untargeted metabolomics methods has enabled the simultaneous, unbiased evaluation of hundreds of metabolites [13].

Therefore, we assessed associations of plasma metabolites identified by untargeted metabolomics with disease severity in an age- and sex-matched case-control study within a genetically homogeneous group of carriers of *MYBPC3* founder variants.

## 2. Results

Subjects had a median age of 57.3 years (interquartile range 39.0–69.1) and 57.1% were male. Subject characteristics are provided in Table 1. Details on relatedness between subjects and genetic testing are provided in Appendix A. Subjects were recruited from a total of 51 families, with a median of one subject (interquartile range 1–2) per family. Additional (likely) pathogenic variants were ruled out using next-generation sequencing panels in 11 subjects (36.7%) with a severe phenotype and three subjects (10.0%) with no or a mild phenotype.

Outcomes in the severe phenotype group included a maximum wall thickness ≥20 mm in 24 subjects (82.8%), septal reduction therapy in 8 subjects (26.7%), malignant ventricular arrhythmia in 7 subjects (23.3%), and heart failure in 14 subjects (48.3%), including congestive heart failure occurred in 10 subjects and systolic dysfunction in 9 subjects. In the no or mild phenotype group, nine subjects fulfilled HCM diagnostic criteria (three with a maximum wall thickness ≥15 mm and six with a maximum wall thickness of 13–14 mm). The distribution of outcomes is shown in Appendix A.

Correlations between the 1903 peaks identified by metabolomics are shown in Appendix A. Although the overall median correlation among peaks was low at 0.089 (IQR 0.042–0.15), the median of the maximum (absolute) correlation was 0.45 (IQR 0.41–0.56), and there was a strong correlation (Spearman’s ρ > 0.8 or <−0.8) among 129 peaks.

### 2.1. Biomarker Identification

The sPLS-DA model identified one component encompassing 50 peaks. The XGBoost model incorporated 146 peaks. Lasso logistic regression selected 11 peaks. The full lists of included peaks are provided in Appendix A. Figure 1 shows the relative importance of the top 25 peaks of each model. Twenty peaks had only one metabolite annotated to them and 22 peaks had multiple metabolites annotated to them.

As shown in Appendix A, none of these 42 top peaks correlated strongly with one another (Spearman’s ρ > 0.8 or <−0.8). Moderate correlations (Spearman’s ρ > 0.5 or <−0.5) were found among 15 peaks, with the highest correlation between the [1,3-dimethyluracil, imidazolepropionic acid and {Pi-}methylimidazoleacetic acid] peak and 3-Methylhistidine (Spearman’s ρ 0.69, *p* < 0.001).

Out of the 42 top peaks, 36 associated with severity with a *p* < 0.05, of which 20 were at *p* < 0.01, with aminoadipic acid, the [2-methoxy-estradiol-17b 3-glucuronide and 4-Hydroxyandrostenodione glucuronide] peak, and the [1,3-dimethyluracil, imidazolepropionic acid and {Pi-}methylimidazoleacetic acid] peak associated at *p* < 0.001. Box plots are provided in Appendix A.

Comparing subjects with no or a mild phenotype to genotype-negative subjects, five peaks were associated at *p* < 0.05, all of which were also associated with HCM severity. The [aspartyl-{iso}leucine, {gamma-}glutamylvaline, and {iso}leucyl-aspartate] peak was associated at *p* < 0.01. Comparing subjects with a severe phenotype to genotype-negative subjects, five peaks were associated at *p* < 0.05 and the [{3-}oxoglutaric acid] peak was associated at *p* < 0.01.

The pathways connected to the top peaks are detailed in Figure 2. Acylcarnitines, histidine metabolism, lysine metabolism, proteolysis, and purine metabolism were each connected to ≥2 metabolites associated at *p* < 0.05, including ≥1 metabolite at *p* < 0.01. Alanine, aspartate, and glutamate metabolism likewise fulfilled the above criteria, both metabolites in this pathway were annotated to peaks that included metabolites from other pathways.

### 2.2. Sensitivity Analyses and External Validation

After excluding 15 subjects with heart failure, Lasso logistic regression included 20 of the 42 top metabolite peaks. Exclusion of eight subjects with prior septal reduction therapy resulted in the Lasso model including 19 of the top metabolite peaks, including 12 retained by the sensitivity analysis excluding subjects with heart failure. Odds ratios for the Lasso logistic regression models are provided in Appendix A. Box plots of metabolite levels are provided in Appendix A.

In the external validation cohort, Lasso logistic regression selected age, sex, and six out of our top ten metabolites, including the [Threoninyl-Tryptophan, Tryptophyl-Threonine] peak, the [2-Methoxy-estradiol-17b 3-glucuronide and 4-hydroxyandrostenedione glucuronide] peak, menadiol dissucinate, aminoadipic acid, the [1,3-Dimethyluracil, imidazolepropionic acid and (Pi-)methylimidazoleacetic acid] peak, 3-fumarylpyruvate, and 9,12-Hexadecadienoylcarnitine. Odds ratios are provided in Appendix A.

## 3. Discussion

In this exploratory age- and sex-matched case-control study, we assessed associations of plasma metabolites with clinical severity of carriers of Dutch *MYBPC3* founder variants, comparing 30 severely affected subjects to 30 subjects with no or only a mild phenotype. Using untargeted metabolomics, we assessed a wide array of metabolites. Combining the top 25 peaks of three supervised models identified 42 candidate peaks, of which 36 were associated with a severe phenotype, with 20 at *p* < 0.01 and 3 at *p* < 0.001. The metabolites annotated to the candidate peaks clustered to several pathways, including acylcarnitines, histidine metabolism, lysine metabolism, proteolysis, purine metabolism, and steroid hormone metabolism.

### 3.1. Energy Metabolism

Many of the peaks identified by our study clustered to pathways involved in energy metabolism. Myocardial proteomics and imaging studies have previously revealed perturbations in the myocardial efficiency and energy metabolism in HCM [9,10,12].

A non-failing heart preferentially utilises fatty acids as its energy source, but in HCM, a shift towards glucose utilisation occurs [15]. Acylcarnitines transport long-chain fatty acids into mitochondria for β-oxidation, thereby reflecting the cardiomyocyte utilisation of fatty acids [16]. Perturbations in acylcarnitines have been reported in HCM [17] and related to disease severity in dilated cardiomyopathy [18]. Conversely, inborn errors of acylcarnitine metabolism, particularly very long-chain acyl-coenzyme A dehydrogenase deficiency, are known to cause HCM phenotypes [19]. Our study likewise identified changes in acylcarnitines as a marker of HCM severity, particularly in 9,12-hexadecadienoyl- (C16:2, *p* = 0.006), 2-octenoyl- (C18:1, *p* = 0.036), and arachidyl carnitine (C20:0, *p* = 0.026).

Additionally, 4-Trimethylammoniobutanal, a product of lysine degradation used in carnitine biosynthesis, was found to be increased in severe HCM subjects (*p* = 0.034). Another lysine metabolite, aminoadipic acid (also known as 2-Aminoadipate), was strongly increased (*p* < 0.001) in severe HCM subjects. Aminoadipic acid was previously associated with cardiac remodelling in the Framingham Study [20] and identified as a biomarker for diabetes risk [21].

### 3.2. Other Pathways

In our study, both uric acid and a peak including 1,3-dimethyluracil, a methyl derivative of uric acid, were associated with HCM severity (*p* = 0.010 and *p* < 0.001, respectively). Uric acid was previously shown to predict heart failure, ventricular arrhythmia, and all-cause mortality in HCM subjects [22,23]. Additionally, increased uric acid levels have been related to myocardial ischemia and heart failure [24].

Several histidine metabolites were associated with HCM severity in our study, i.e., 3-methylhistidine (*p* = 0.018), 2-(3-Carboxy-3-(methylammonio)propyl)-L-histidine (*p* = 0.016), hydantoin-5-propionic acid (*p* = 0.009), and a peak including imidazolepropionic acid and methylimidazoleacetic acid (*p* < 0.001; the same peak as 1,3-dimethyluracil). 3-methylhistidine acts as a marker for myofibrillar breakdown. It was identified as a prognostic marker in heart failure [25] and associated with severity in dilated cardiomyopathy [18]. 2-(3-Carboxy-3-(methylammonio)propyl)-L-histidine is a modified histidine residue of elongation factor 2, an essential factor for protein synthesis, which was found to be overexpressed in end-stage heart failure HCM patients [26]. Methylimidazoleacetic acid is the main metabolite of histamine. Histamine-related mechanisms have been suggested in heart failure. The use of histamine H_2_ receptor antagonists was associated with preserved left ventricular morphological indices and a lower risk of incident heart failure [27]; likewise, a small (*n* = 50) randomised controlled trial in heart failure patients found that famotidine, an H_2_ receptor antagonist, improved heart failure symptoms and decreased heart failure readmissions [28]. To the best of our knowledge, no studies have related hydantoin-5-propionic acid, or imidazolepropionic acid, a microbially produced histidine metabolite that impairs insulin signalling [29], to HCM or heart failure.

One peak that was negatively associated with HCM severity (*p* < 0.001) was paradoxically annotated to both an oestrogen metabolite, 2-Methoxy-estradiol-17b 3-glucuronide, and an aromatase inhibitor metabolite, 4-hydroxyandrostenedione glucuronide. The effects of sex hormones on HCM pathogenesis are currently being debated as fewer females are diagnosed with HCM, albeit often at an older age and with worse symptoms and prognosis [30]. A murine HCM model indicated a protective effect of oestrogen [31]; however, the effects in humans remain unclear. Still, the observation in our study warrants further studies into the role of sex hormones in HCM.

Multiple di- and oligopeptides connected to proteolysis were identified in our study. Previous studies have suggested stress on protein quality control systems as one of the mechanisms behind *MYBPC3* haploinsufficiency-mediated HCM [32,33], which could lead to a build-up of products of incomplete proteolysis. However, some of the di-/oligopeptides in our study were significantly increased (e.g., Hydroxyprolyl-Histidine/Histidylhydroxyproline, *p* = 0.005), whereas others were significantly decreased in severe HCM (e.g., Threoninyl-Tryptophan/Tryptophyl-Threonine, *p* = 0.002). Therefore, it remains unclear whether deficient protein quality control systems or rather metabolite-specific changes underlie these findings.

### 3.3. Previous Metabolomics Studies

To the best of our knowledge, our study is the first to comprehensively compare metabolites among severely and mildly affected carriers of HCM-causing variants while accounting for confounding from age and sex. Five previous studies assessed blood-based metabolites using metabolomics in HCM patients, each with methodological differences compared with our study [34,35,36,37,38]. These include comparisons to hospital [35] or general population controls [36], not matching on age [37] or sex [36,37], differences in the range of analysed metabolites [34,35,36,38], and differences in outcomes [34,35,36,37,38].

Nevertheless, taken together, these studies indicate pathways involved in energy metabolism including acylcarnitines; eicosanoids; proteolysis products; and metabolites from histidine, lysine, purine, and steroid hormone metabolism. Importantly, proteomics and multi-omics studies in tissue of HCM patients likewise demonstrated perturbations in acylcarnitines and histidine, lysine, and purine metabolites [10,11,12], corroborating our findings.

### 3.4. Limitations and Future Directions

The limited sample size of this study precluded analyses for specific heart failure, ventricular arrhythmia, and LVOT obstruction outcomes, as well as correction for potential confounders including medication usage and kidney function or known prognostic factors. Furthermore, the cross-sectional case-control design of this study limits causal and prognostic inference. Large prospective studies are required to assess the incremental prognostic utility of the metabolites indicated by our study for specific outcomes. Randomised-controlled trials, Mendelian randomisation studies, and experimental studies are required to assess the causality of the indicated pathways.

Although all carriers of each founder variant are by extension related to one another, differing degrees in relatedness and concurrent genetic background may confound the associations between HCM severity and metabolites. This confounding can be ameliorated by matching or stratifying by relatedness. In our study, the use of age- and sex-matching precluded matching by relatedness. Instead, relatedness between the subjects in our study was limited, which should likewise reduce confounding from genetic background. Moreover, metabolic perturbations that result from differences in genetic background may still be clinically relevant, as long as the genetic variability and corresponding metabolic differences are present in the broader (HCM) population. Further studies are required to assess the interplay between metabolites, genetic background, and HCM severity.

Our study included a relatively large number of subjects with end-stage HCM, characterised by congestive heart failure and/or systolic dysfunction. Likewise, several subjects had previously undergone septal reduction therapy, which may alter cardiac metabolism [9]. Additionally, we only selected carriers of *MYBPC3* variants, which may limit generalisability to other genotypes. Furthermore, only a selection of our subjects underwent genetic testing using next-gen sequencing approaches. Therefore, the effects of additional genetic variants cannot be ruled out. Still, several of the metabolites indicated by our study retained their associations after exclusion of subjects with heart failure or prior septal reduction therapy, and six out of the top ten metabolites were externally validated in patients with LVOT obstruction with diverse genotypes.

Our study utilised untargeted metabolomics, a non-quantitative method with high costs and workload. Therefore, our results require validation using robust, easy-to-perform, and cheaper quantitative methods. Additionally, we did not correct for multiple testing as this is an exploratory study, which increases the risk of type I error. Therefore, confirmatory studies using highly specific quantification methods are required. Furthermore, we only measured metabolites in plasma. Recent data indicate that extracardiac production of several novel heart failure biomarkers strongly influences their plasma levels [39]. We cannot exclude extracardiac production of biomarkers identified by our study.

Finally, blood samples were not obtained under specific conditions, e.g., in a fasting state or after exercise. This likely increased the variability in metabolite measurements, which may have generated false-positive results or prevented the identification of potential biomarkers. However, an ideal biomarker would be predictive independent of such sampling conditions or other factors affecting its biological variability [40].

## 4. Materials and Methods

### 4.1. Subject Inclusion

This study consisted of an exploratory, nested case-control study within the *BIO FOr CARe* (Identification of biomarkers of hypertrophic cardiomyopathy development and progression in Dutch *MYBPC3*
founder variant carriers) cohort study [41]. In summary, carriers of the *MYBPC3* c.2373dupG, c.2827C > T, c.2864_2865delCT, or c.3776delA variants aged ≥18 years were included from January 2017 onwards to prospectively undergo blood collection.

Thirty individuals with a severe phenotype were age- and sex-matched to 30 with no or a mild phenotype (i.e., not fulfilling criteria for a severe phenotype). A severe phenotype was defined as a documented maximum wall thickness ≥20 mm, LVOT obstruction necessitating septal reduction therapy, occurrence of heart failure (congestive heart failure or systolic dysfunction, defined as a left ventricular ejection fraction <50%), or malignant ventricular arrhythmia (sustained ventricular tachycardia, i.e., >30 s, with haemodynamic instability or requiring earlier termination, ventricular fibrillation, appropriate implantable cardioverter-defibrillator intervention, or resuscitated cardiac arrest). Patients with severe liver or kidney failure were excluded. Additionally, 10 age- and sex-matched genotype-negative family members were included as genotype-negative controls.

This study was performed in accordance with the Helsinki declaration and was approved by the Medical Ethics Committee of the UMC Utrecht. Written informed consent was obtained from all subjects.

### 4.2. Metabolomics

Venous blood samples were collected under non-fasting, resting conditions in heparin-containing tubes and processed within 45 min in accordance with our previously published protocol [41]. Samples were stored at −80 °C until analysis. All samples were analysed as a single batch, using an untargeted direct-infusion high resolution mass spectrometry and metabolite identification method [42]. In short, this method consistently and accurately identified 1903 mass peaks corresponding to 3904 metabolite annotations (including isomers).

### 4.3. Biomarker Identification

Three distinct supervised methods that use regularisation to reduce overfitting and allow feature selection were fitted to the metabolomics data of subjects with severe phenotypes and subjects with no or mild phenotype. Sparse partial least squares discriminant analysis (sPLS-DA), a dimensionality reduction method, was performed on scaled and centred peak intensities and tuned using fivefold cross-validation repeated 50 times using the “mixOmics” package [43]. Gradient boosting was performed on crude peak intensities using XGBoost, which is able to detect non-linear interactions between metabolites and outcomes by utilising gradient boosted decision trees [44]. Tuning was performed using grid searches with fivefold cross-validation using the “caret” package [45]. Lasso logistic regression was performed as a more conventional feature selection method, using scaled and centred peak intensities, with lambda determined using fivefold cross-validation, using the “glmnet” package [46]. Top peaks were selected based on the absolute values of weight coefficients in sPLS-DA, the gain metric in XGBoost, and the absolute values of regression coefficients in Lasso logistic regression. As a secondary analysis to aid the interpretation of the results of these models, associations of the top 25 peaks of each model with HCM severity were further explored using Mann–Whitney U and Kruskal–Wallis tests.

As peaks are annotated solely on accurate mass, identification is putative and peaks may have multiple metabolites annotated to them, hereafter described as peaks with annotated metabolites in square brackets (“[ ]”). Pathway analysis was performed to cluster the peaks to pathways that may be associated with severe HCM and derive the more likely involved metabolites. The metabolites annotated to the top 25 peaks identified by each of the sPLS-DA, XGBoost, and Lasso logistic regression models were assessed in the Human Metabolome Database [47], the Kyoto Encyclopaedia of Genes and Genomes [48] pathway database, and by manual reference searching.

### 4.4. Statistical Analysis

Dichotomous and categorical variables are presented as counts with percentages and were analysed using two-sided Fisher’s exact test. Continuous variables are presented as means ± standard deviations or medians with interquartile ranges (IQR), according to their distribution. Normally distributed variables were analysed using the unpaired t-test or one-way ANOVA and non-normally distributed variables were analysed using the Mann–Whitney U or Kruskal–Wallis test. Correlations between metabolites were calculated using Spearman’s ρ and visualised as a heatmap with the “pheatmap” package [49]. All analyses were conducted in *R* version 4.1.2 (R Development Core Team, Vienna, Austria, 2021) using *RStudio Desktop* version 2021.09.1+372 (RStudio Team, Boston, Massachusetts, United States of America, 2021).

### 4.5. Sensitivity Analyses and External Validation

To assess the influence of heart failure on our results, we performed a sensitivity analysis in subjects without congestive heart failure or systolic dysfunction and matched subjects with no or a mild phenotype. Lasso logistic regression was performed using fivefold cross-validation to assess the top metabolites identified by the sPLS-DA, XGBoost, and Lasso logistic regression performed on all subjects, and associations between subjects with severe phenotypes and those with no or a mild phenotype were assessed using Mann–Whitney U. Similarly, to assess the effects of septal reduction therapy, we performed a sensitivity analysis without septal reduction therapy prior to blood collection.

To assess the reproducibility of our results and assess the effects of fasting, we performed external replication on 14 subjects with symptomatic LVOT obstruction and 31 unmatched, asymptomatic carriers of (likely) pathogenic variants from the previously published Engine study [37]. The top 10 peaks from the present study were evaluated using Lasso logistic regression using fivefold cross-validation, additionally including age and sex as covariates.

## 5. Conclusions

This exploratory case-control study comprehensively analysed metabolites in carriers of Dutch *MYBPC3* founder variants with severe HCM phenotypes and age- and sex-matched carriers with no or only a mild phenotype. This revealed multiple potential biomarkers associated with disease severity, suggesting that several pathways, including acylcarnitine, histidine, lysine, purine, and steroid hormone metabolism, as well as proteolysis, are dysregulated in patients with severe phenotypes. Further studies are required to replicate these findings using quantitative methods, determine causality, and assess whether the biomarkers further improve risk stratification.

## Figures and Tables

**Figure 1 ijms-24-04031-f001:**
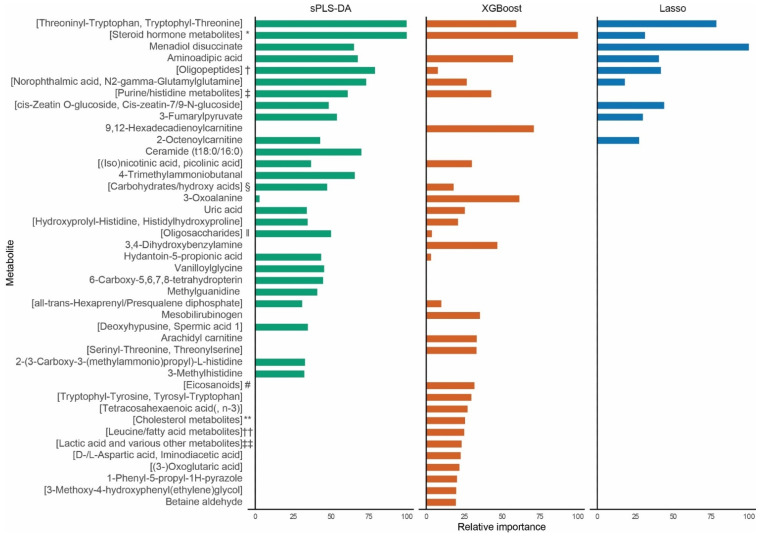
Relative variable importance. Top 25 metabolites in each model ordered by their mean relative importance across the models, showing their relative importance scaled to the most important metabolite in each model. The sPLS-DA and XGBoost models show 26 and 30 metabolites, respectively, as these metabolites were included in the top 25 of other models. Two metabolites selected by the Lasso logistic regression, Ceramide (t18:0/16:0) and 4-Trimethylammoniobutanal, are not visible owing to their very small coefficients (−8.67 × 10^−7^ and 1.82 × 10^−6^, respectively). Metabolites annotated to the same mass spectrometry peak are denoted in square brackets. sPLS-DA, sparse partial least squares discriminant analysis. * includes 2-methoxy-estradiol-17b 3-glucuronide and 4-hydroxyandrostenedione glucuronide. † includes aspartyl-(iso)leucine, L-beta-aspartyl-L-leucine, (gamma-)glutamylvaline, and (iso)leucyl-aspartate. ‡ includes 1,3-dimethyluracil, imidazolepropionic acid, and (Pi-)methylimidazoleacetic acid. § includes (S)-3,4-/2,4-dihydroxybutyric acid, 4-deoxythreonic/-erythronic acid, erythrose, and L-erythrulose. ‖ includes dextrin, D-Gal alpha 1- > 6D-Gal alpha 1- > 6D-glucose, 3-galactosyllactose, 1-kestose, maltotriose, and melezitose. # includes 5,6-/8,9-/11,12-/14,15-dihydroxyeicosatrienoic acid, 6,7-dihydro-12-epi-/10,11-dihydro-/12-keto-tetrahydro-leukotriene B4, and 15-hydroperoxyeicosa-8Z,11Z,13E-trienoate. ** includes cholestane-3b,5a,6b-triol and 5b-cholestane-3a,7a,12a-triol/-3a,7a,26-triol/-3a,7a,27-triol. †† includes hexanoylglycine, isovalerylalanine/-sarcosine, and N-acetylleucine. ‡‡ includes D-/L-lactic acid, hydroxypropionic acid, glyceraldehyde, dihydroxyacetone, and methoxyacetic acid.

**Figure 2 ijms-24-04031-f002:**
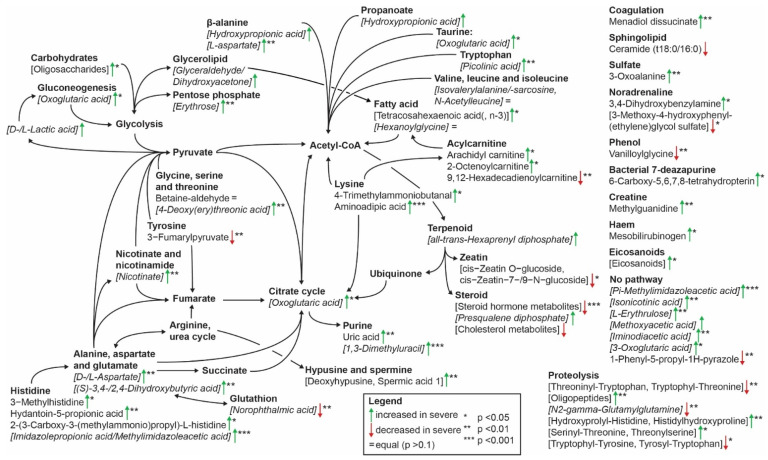
Pathway analysis in peaks with multiple annotated metabolites. Pathways identified for the top 25 metabolites in each of the biomarker identification models. Square brackets indicate metabolites annotated to peaks including other metabolites; italics indicate that ≥1 annotated metabolite was connected to ≥1 different pathway. Arrows indicate metabolites increased (green upward arrow) or decreased (red downward arrow) in subjects with severe phenotypes compared with subjects with no or mild phenotypes. Asterisks denote the level of significance in Mann–Whitney U tests (* *p* < 0.05, ** *p* < 0.01, *** *p* < 0.001). Metabolites that did not associate with disease severity (*p* > 0.1) are indicated with an equals sign (“=”). Lists of annotated metabolites were condensed as in Figure 1.

**Table 1 ijms-24-04031-t001:** Subject characteristics. Data shown as counts (%), means (standard deviation), or medians [interquartile range]. *p*-values are provided for comparisons across groups for age and sex and for comparisons between the severe and no/mild phenotype groups for the remaining variables; *p*-values < 0.05 are highlighted in bold. Family history of sudden cardiac death was defined in accordance with the HCM-Risk SCD calculator [14]. SCD, sudden cardiac death; VT, ventricular tachycardia; LVOT, left ventricular outflow tract; LV, left ventricular.

	Severe Phenotype	No/Mild Phenotype	Genotype-Negative	*p*-Value
	(*n* = 30)	(*n* = 30)	(*n* = 10)	
Age (years)	56.3 (38.2–71.7)	58.4 (38.4–68.4)	55.5 (41.5–64.5)	0.835
Male sex	17 (56.7)	17 (56.7)	6 (60.0)	1.00
Index patient	18 (60.0)	5 (16.7)		**0.001**
Body surface area (m^2^)	2.0 (1.8–2.1)	1.9 (1.8–2.1)		0.437
Syncope	8 (27.6)	2 (6.7)		**0.042**
Family history of SCD	10 (37.0)	7 (25.9)		0.559
Non-sustained VT	16 (61.5)	8 (42.1)		0.237
Maximum wall thickness (mm)	21 (16–23)	11 (9–13)		**<0.001**
Left atrial diameter (mm)	43 (40–50)	37 (35–42)		0.062
LVOT gradient (mmHg)	4 (3–80)	5 (4–6)		0.721
LV ejection fraction (%)	58 (50–60)	60 (58–65)		**0.010**
Atrial fibrillation	12 (41.4)	2 (6.7)		**0.002**

## Data Availability

Data supporting the reported results are available upon request.

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
