# Peer review of "Untargeted Metabolomics Identifies Potential Hypertrophic Cardiomyopathy Biomarkers in Carriers of MYBPC3 Founder Variants"

_ijms, 2023, doi:10.3390/ijms24044031_

Round 1
Reviewer 1 Report
The authors present a metabolomic study in hypertrophic cardiomyopathy (HCM). This approach is in "hot topics' of research interest. They use the case-control design of the study and found a set of metabolites associated with severe phenotype of the HCM.
I have minor comments:
1) I would appreciate if the author describe how they performed genotyping for the identification of variant carriers (NGS, Sanger sequencing?)
2) What clinical perspectives do metabolic studies in HCM have?
Author Response
We thank the reviewer for their comments and are glad to hear they share our interest in this approach.
1) I would appreciate if the author describe how they performed genotyping for the identification of variant carriers (NGS, Sanger sequencing?)
We have added a table to the supplement (Table S1) detailing the genotyping that was performed in the subjects, and shortly summarised this in the results section ("Additional (likely) pathogenic ... or mild phenotype.", page 2, lines 80-83).
2) What clinical perspectives do metabolic studies in HCM have?
Clinically, metabolic studies in HCM may identify metabolites that could serve as prognostic markers and identify pathways involved in HCM pathogenesis that could potentially be of interest as treatment targets. We have added a sentence to the introduction to emphasise these applications ("This may lead ... potential treatment targets", page 2, lines 69-71). The further steps required to reach these ends are already discussed in our limitations section.

Reviewer 2 Report
This study enters the new and exciting field of exploring the boundaries of the predictive values and applicability of metabolic studies for HCM .
Among the concerns over the current study is the statistical power given the small sample size and large number of variables that were tested.
Q: was N metabolites 3904 as on page 2 line 85 or 3835 as page 8 line 309. Please provide these power calculation and include the effect that correcting for multiple testing.
The novelty of this type of association studies evoked other specific questions regarding the study design.
As the authors correctly state, the reason this study was performed is that HCM, like most other common genetic disorders is less monogenic that we previously believed and additional risk factors determine the outcome: there is a great variation in disease expression among carriers of founder MYBCP3 variants, classified as (likely)pathogenic, including a relatively large proportion in which the defect is non penetrant, a proportion with mild features and some showing severe cardiac remodeling.
Thus we want to know more about the additional risk factors that play an important role in the development of HCM features, and the severity of the outcome. For now, we do not know which combination of risk factors is necessary to instigate fibrosis and hypertrophy and why this process is so different between carriers of similar genetic defects. Here metabolomics are used to detect the additional effect that may attenuate or enforce the underlying founder variants in MYBPC3.
Comparing relatives within families with similar and different phenotypes is a powerful method to detect or reject underlying genetic modifiers and their translated effects measured by metabolomics. It is an omission of the current study not to use family segregation data. Given that the study population consist of index cases and affected and unaffected carriers and healthy relatives without the familial founder variant of carriers.
To identify substrates associated with severe phenotypes, the metabolic overlap and discrepancies between relatives with severe HCM and relatives with mild symptoms can be analyzed and presented separately as well as the metabolic overlap between the non carriers and the carriers. In this way, more accurate predictors of severe phenotypes and for effect modifiers can found and false associations excluded.
Q; provide an additional overview with a stratification of the metabolites according to familial segregation of phenotype, including the genotype neg.
The box plots in the supplement show that the genotype negatives and the severe phenotypes are significantly different for only for a small number of the identified top 25 metabolites. This would make it less likely that these metabolites are associated with severe cardiac disease.
Q: the question is if Figure 1 with the overview of relative importance of metabolites, is a overrepresentation and should be adjusted to contain only the metabolites that are significantly different between severe and mild cases, the selection that is significantly different between severe and genotype negatives, and the selection that is significantly different between mild and genotype negative.
Author Response
We are grateful to the reviewer for their comments and glad to hear that they share our excitement for this topic.
Q: was N metabolites 3904 as on page 2 line 85 or 3835 as page 8 line 309. Please provide these power calculation and include the effect that correcting for multiple testing.
The novelty of this type of association studies evoked other specific questions regarding the study design.
A: We apologise for the confusion. The numbers listed on page 8 were taken from a previously published article reporting the metabolomics method in different types of samples (plasma, dried blood spot). We have amended the sentence on page 8 to show the number of peaks and annotated metabolites identified in plasma in our study (“In short, this … annotations (including isomers).”). To avoid confusion, we have also amended the section on page 2 to avoid duplicating part of the methods in the results section (“Metabolomics identified 1903 … in Figure S2.”).
The reviewer raises an important point regarding statistical power and multiple testing, as these are clearly an issue when assessing 3904 variables. Therefore, we used methods that use regularisation to perform feature selection and reduce overfitting. However, as this is an exploratory study, we chose not to perform post-selection statistical inference analyses. The p-values from Mann-Whitney U tests in Figure 2 and boxplots in the supplement are intended to aid interpretation of these results. This has been emphasised in the methods section of the manuscript ("As a secondary ... and of these models,", page 9, lines 342-343).
As stated in the limitations & future directions section (page 8, lines 288-289), this approach does increase the risk of type I error. In any case, we feel that non-quantitative approaches such as untargeted metabolomics require validation using quantitative methods, and these confirmatory studies will deal with type I error in our exploratory study.
Q; provide an additional overview with a stratification of the metabolites according to familial segregation of phenotype, including the genotype neg.
A: We agree with the reviewer that a study that stratifies metabolites according to familial segregation of severe and less severe phenotypes would be of great interest to filter out false positive associations and in other ways assess the interplay between genetic background, metabolites and phenotypic severity. However, as a result from the age- and sex-matching, only a small number of related subjects were included in our study, as indicated by a median number of subjects per family of one (interquartile range 1-2). We have added an overview of the subjects per family to Table S1 in the supplement, and have added mention of the number of subjects per family in the manuscript ("Details on relatedness... 1-2) per family.", page 2, lines 78-80).
Therefore, stratification according to relatedness would not be feasible in our study. We have added this to our limitations section ("Although all carriers ... and HCM severity.", page 7, lines 265-275). Of note, the limited relatedness between subjects in our study should also reduce potential confounding from genetic background.
Q: the question is if Figure 1 with the overview of relative importance of metabolites, is a overrepresentation and should be adjusted to contain only the metabolites that are significantly different between severe and mild cases, the selection that is significantly different between severe and genotype negatives, and the selection that is significantly different between mild and genotype negative.
A: As mentioned in our answer to the first question, the boxplots with Mann-Whitney U and Kruskal-Wallis tests are meant to illustrate differences in metabolites, but not to replace the primary analysis consisting of the sPLS-DA, XGBoost and Lasso logistic regression. In particular, the XGBoost is able to capture non-linear interactions between variables and the outcome, which the boxplots would fail to detect. Therefore, filtering the metabolites listed in Figure 1 based on the Mann-Whitney U and Kruskal-Wallis tests would result in the loss of potentially interesting metabolites. To emphasise this, we have elaborated on the specific characteristics of each of the three models in our primary analysis in our methods section ("Three distinct supervised ... the "glmnet" package [47].", page 9, lines 329-340).
We feel that the overrepresentation that the reviewer cautions us for is already mitigated in two ways. Firstly, we show the variable importance for metabolites identified by sPLS-DA, XGBoost and Lasso in separate columns, so that those more interested in the results from the more conventional Lasso method may opt to focus solely on that column. Secondly, the significance levels from the Mann-Whitney U analyses are included in the pathway analysis in Figure 2, allowing readers to discern the metabolites with more evident differences in boxplots (e.g. p<0.01 or p<0.001). This way we hoped to provide a balanced representation between the results from the more novel models and the more conventional methods.